# Osteoblast-intrinsic defect in glucose metabolism impairs bone formation in type II diabetic male mice

**Fangfang Song[1,2], Won Dong Lee[3], Tyler Marmo[1], Xing Ji[1], Chao Song[1], Xueyang Liao[1], Rebecca Seeley[1], Lutian Yao[1], Haoran Liu[4], Fanxin Long[1,5]\***

[1]Translational Research Program in Pediatric Orthopedics, Department of Surgery, The Children's Hospital of Philadelphia, Philadelphia, United States; [2]The State Key Laboratory Breeding Base of Basic Science of Stomatology and Key Laboratory for Oral Biomedicine of Ministry of Education, School and Hospital of Stomatology, Wuhan University, Wuhan, China; [3]Lewis Sigler Institute for Integrative Genomics, Princeton University, Princeton, United States; [4]Department of Computer Science, New Jersey Institute of Technology, Newark, United States; [5]Deaprtment of Orthopedic Surgery, University of Pennsylvania, Philadelphia, United States

**Abstract** Skeletal fragility is associated with type 2 diabetes mellitus (T2D), but the underlying mechanism is not well understood. Here, in a mouse model for youth-onset T2D, we show that both trabecular and cortical bone mass is reduced due to diminished osteoblast activity. Stable isotope tracing in vivo with $^{13}$C-glucose demonstrates that both glycolysis and glucose fueling of the TCA cycle are impaired in diabetic bones. Similarly, Seahorse assays show suppression of both glycolysis and oxidative phosphorylation by diabetes in bone marrow mesenchymal cells as a whole, whereas single-cell RNA sequencing reveals distinct modes of metabolic dysregulation among the subpopulations. Metformin not only promotes glycolysis and osteoblast differentiation in vitro, but also improves bone mass in diabetic mice. Finally, osteoblast-specific overexpression of either Hif1a, a general inducer of glycolysis, or Pfkfb3 which stimulates a specific step in glycolysis, averts bone loss in T2D mice. The study identifies osteoblast-intrinsic defects in glucose metabolism as an underlying cause of diabetic osteopenia, which may be targeted therapeutically.

**\*For correspondence:** longf1@chop.edu

**Competing interest:** The authors declare that no competing interests exist.

## Editor's evaluation

The authors have established an important mouse model for youth-onset T2D and provided convincing evidence to demonstrate that suppression of glucose metabolism in osteoblasts is the main cause of impaired bone formation in T2D. Particularly, the authors uncovered that activation of glycolysis in osteoblasts restores bone formation in T2D mice as an essential mechanism. This study fundamentally demonstrated osteoblast-intrinsic defects in glucose metabolism as an underlying cause of diabetic osteopenia with therapeutical potential.

## Introduction

Bone quality is maintained by the coordinated functions of osteoblasts and osteoclasts. During bone formation, polarized osteoblasts secrete extracellular matrix (ECM), consisting of type I collagen and non-collagenous matrix proteins, that mineralize over time (*Eriksen, 2010*; *Long, 2012*). The production of ECM by osteoblasts is an energy-demanding process and likely requires metabolic reprogramming, but the metabolic regulation of osteoblast differentiation and function is largely unknown

(*Buttgereit and Brand, 1995*; *Rolfe and Brown, 1997*; *Lee et al., 2017*). Interestingly, prominent signaling pathways that promote or suppress osteogenesis alter glucose metabolism. For instance, parathyroid hormone (PTH), Wnt, insulin-like growth factors (Igf), and Hif1a stimulate glycolysis while promoting osteogenesis, whereas Notch signaling suppresses glycolysis and osteoblast differentiation in mesenchymal progenitors (*Esen et al., 2013*; *Esen et al., 2015*; *Chen et al., 2019*; *Regan et al., 2014*; *Lee and Long, 2018*). In addition, a number of studies have identified glucose as the main energy substrate for osteoblasts (*Wei et al., 2015*; *Zoch et al., 2016*; *Lee et al., 2020*; *Guntur et al., 2014*). Thus, accumulating evidence to date supports an important role of glucose metabolism in the regulation of osteoblast development and function.

T2D is characterized by insulin insensitivity in target tissues resulting in hyperglycemia. Adults with T2D have increased fracture risk despite having normal or increased areal bone mineral density (aBMD) (*Bonds et al., 2006*; *Ma et al., 2012*; *Sellmeyer et al., 2016*). The poor bone quality has been often attributed to abnormal accumulation of advanced glycation end products in the bone matrix, but could also result from reduced bone turnover as indicated by suppression of both bone resorption and formation in T2D (*Farr and Khosla, 2016*; *Li et al., 2022*; *Moseley et al., 2021*). Youth-onset T2D has emerged as a significant and increasing health burden in adolescents and young adults (*Nadeau et al., 2016*). Distinct from adult T2D, T2D in youth has been reported to reduce bone mass due to impaired bone anabolism (*Kindler et al., 2020*). Overall, both types of T2D diminish bone formation, but the underlying mechanisms are not fully understood.

Here, we have established and interrogated a mouse model for youth-onset T2D. We show that suppression of glucose metabolism in osteoblasts is the main cause of impaired bone formation in

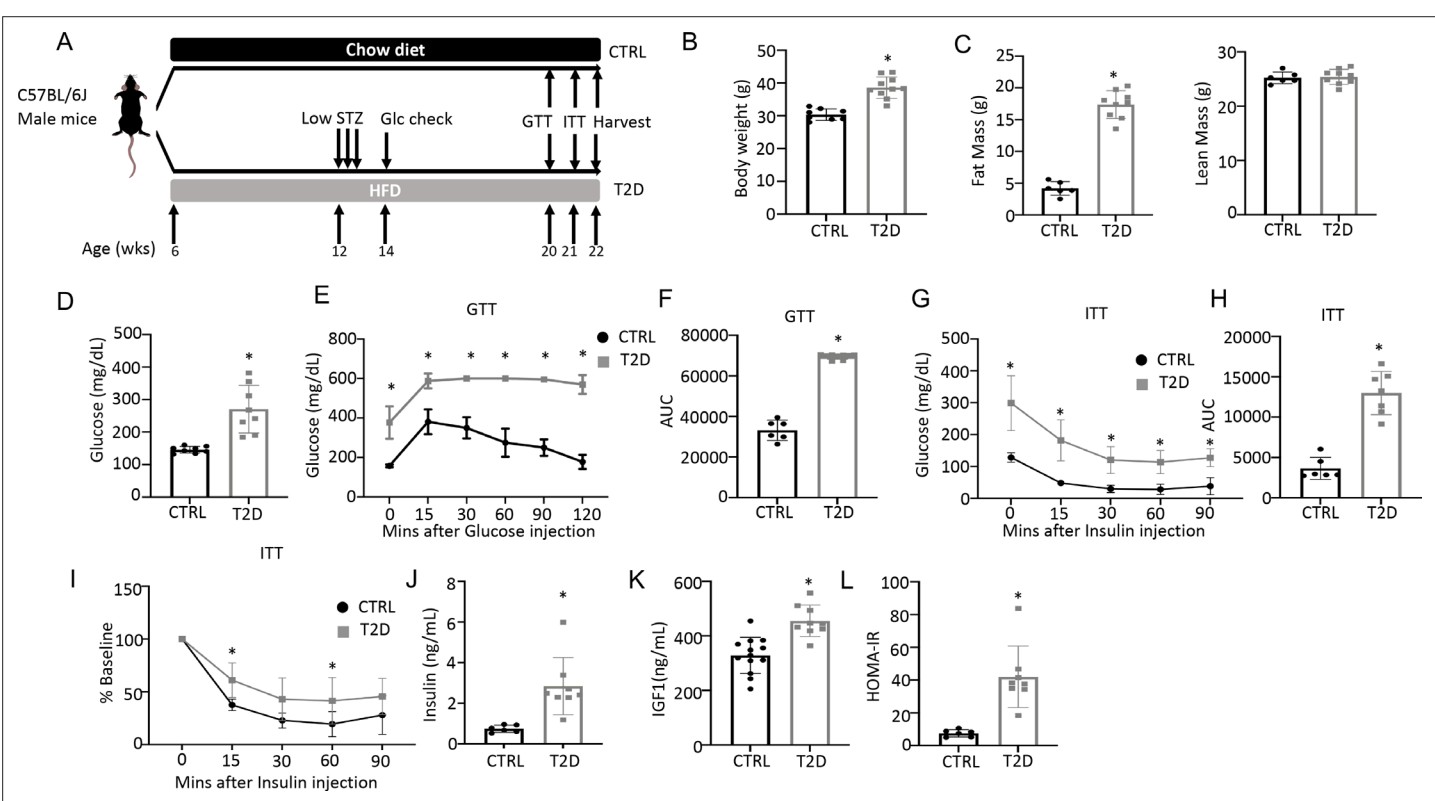

**Figure 1.** Combination of HFD and low-dose streptozotocin (STZ) induces Type 2 diabetes (T2D) in mice. (**A**) A schematic for experimental design. Low-dose STZ: 30 mg/kg body weight. (**B, C**) Body weight (B, CTRL, n=8; T2D, n=10) and body composition by DXA (C, CTRL, n=6; T2D, n=9) at harvest. (**D**) Glucose levels after 6 hr fasting immediately before harvest. n=8. (**E, F**) Glucose tolerance test (GTT) curve (**E**) and area under curve (AUC) (**F**). CTRL, n=6; T2D, n=9. (**G, H**) Insulin tolerance test (ITT) glucose curve (**G**), AUC, (**H**) and percentage of baseline (**I**). CTRL, n=6; T2D, n=7. (**J, K**) Serum insulin (J, CTRL, n=6; T2D, n=8) and Igf1 (K, CTRL, n=9; T2D, n=13) levels. (**L**) HOMA-IR graphs. CTRL, n=6; T2D, n=8. All bar graphs are presented as mean ± SD. unpaired student's t-test (**B–D**, **F**, **H**, **J–L**) or two-way ANOVA with Sidak's multiple comparisons test (**E, G, I**). *p<0.05.

The online version of this article includes the following figure supplement(s) for figure 1:

**Figure supplement 1.** Type 2 diabetes (T2D) mice increase fat mass over lean mass.

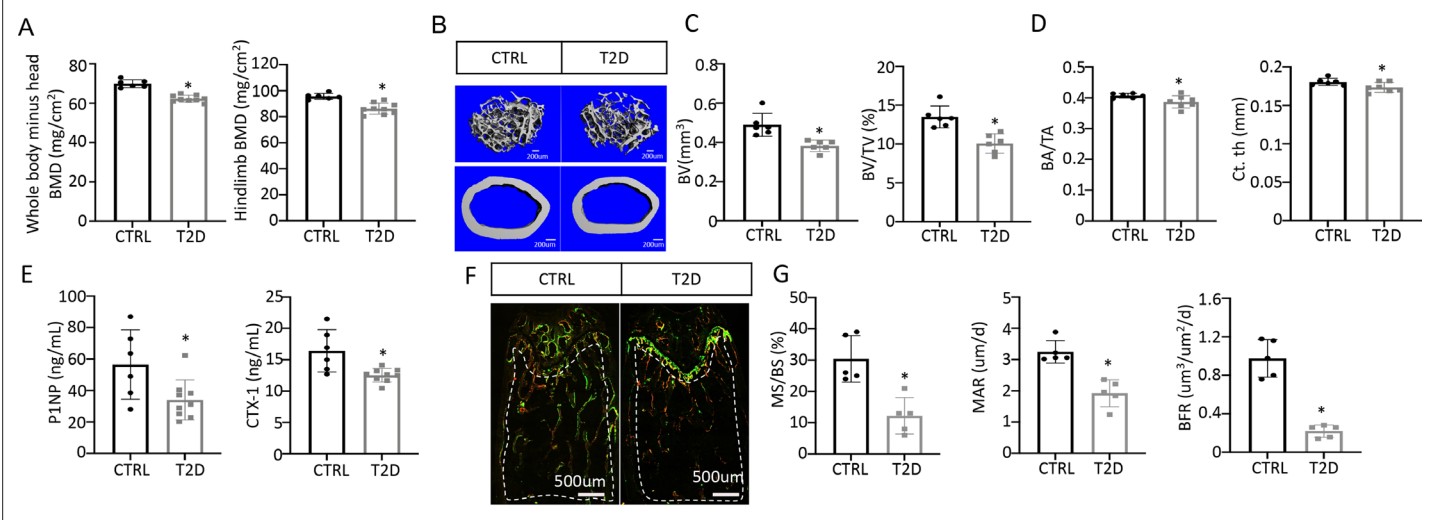

**Figure 2.** Type 2 diabetes (T2D) causes low turnover osteopenia. (**A**) Bone mineral density (BMD) for the whole body (minus head) and one leg (femur and tibia). CTRL, n=6; T2D, n=9. (**B**) Representative µCT images. (**C**) Quantitative analysis of trabecular bone of the distal femur. n=6. (**D**) Quantitative analysis of cortical bone in femur. CTRL, n=6; T2D, n=7. (**E**) Serum bone formation (P1NP) and resorption (CTX-1) markers. CTRL, n=6; T2D, n=9. (**F, G**) Representative images (**F**) and quantification (**G**) for double labeling of trabecular bone in the distal femur. n=5. Data are mean ± SD. Unpaired Student's t-tests. *p<0.05.

The online version of this article includes the following figure supplement(s) for figure 2:

**Figure supplement 1.** Diabetic osteopenia in Type 2 diabetes (T2D) mouse model.

T2D. Importantly, activation of glycolysis in osteoblasts restores bone formation in T2D mice. Together, we identified osteoblast-intrinsic defects in glucose metabolism as an underlying cause of diabetic osteopenia which may be targeted therapeutically.

## Results

### Modeling youth-onset T2D in the mouse

To generate a mouse model for human youth-onset T2D, 6-week-old C57BL/6 J male mice were fed a high-fat diet (HFD) for 6 weeks and then injected with a low dose of streptozotocin (STZ) once daily for three consecutive days, followed by continuous HFD feeding until harvest at 22 weeks of age (*Figure 1A*). Random glucose levels were checked two weeks after the first STZ injection and mice with a glucose level lower than 200 mg/dL were excluded from the study. At harvest, the T2D mice had a significantly higher body weight due to increased fat mass with no change in lean mass (*Figure 1B and C*, *Figure 1—figure supplement 1*). They exhibited notable hyperglycemia and significant impairment in glucose handling when compared to the controls, as revealed by the glucose tolerance test (GTT) and insulin tolerance test (ITT) (*Figure 1D–I*). Fasting serum insulin and Igf1 concentrations were significantly higher in the T2D mice, confirming that they were less sensitive to insulin and reflecting an early phase of T2D (*Figure 1J–K*). Their insulin resistance was further demonstrated by the elevated index of the Homeostatic Model Assessment for Insulin Resistance (HOMA-IR) (*Figure 1L*). Overall, the mouse model exhibits key features of T2D including obesity, hyperglycemia, hyperinsulinemia (early feature), and insulin resistance.

### T2D reduces bone mass by suppressing osteoblast activity

We next examined bone phenotypes in the T2D mice. Dual-energy x-ray absorptiometry (DEXA) at harvest detected a clear decrease in BMD of both the whole body (minus the head) and the hindlimb in comparison with the control (*Figure 2A*). The body length from nose tip to the tail base and the femur length were slightly longer in T2D than normal, perhaps due to increased Igf1 levels (*Figure 2—figure supplement 1A, B*). Imaging and quantification with µCT revealed that T2D significantly reduced trabecular bone volume (BV) and its ratio over tissue volume (BV/TV) in the femur, without altering the tissue volume (TV) itself (*Figure 2B* upper, *Figure 2C*, *Figure 2—figure*

*supplement 1C*). The trabecular bone loss was due to reduced trabecular number (Tb. N) and connectivity (Conn. Dens) concurrent with increased trabecular spacing (Tb. Sp) (*Figure 2—figure supplement 1D–F*). Similarly, in cortical bone, T2D decreased bone area relative to total area (BA/TA) as well as cortical bone thickness (Ct. Th) at the diaphysis of the femur (*Figure 2B* lower, *Figure 2D*, *Figure 2—figure supplement 1G, H*). Thus, like human patients, mice with youth-onset T2D exhibit osteopenia.

We next investigated the cellular basis for diabetic osteopenia. Serum P1NP (procollagen type I N-terminal propeptide) and CTX-1 (collagen type I C- telopeptide) levels were both lower in T2D than in control (*Figure 2E*), indicating that T2D suppressed the overall bone turnover whereas reduced bone formation was responsible for the net bone loss. To further quantify osteoblast activity, we performed dynamic histomorphometry with dual dyes (calcein followed by alizarin red) that labeled active bone-forming surfaces (*Figure 2F–G*, *Figure 2—figure supplement 1I–L*). Quantification within the trabecular bone region of the distal femur detected a marked decrease in both mineralizing surface relative to bone surface (MS/BS) and the mineralization apposition rate (MAR), resulting in an 80% suppression of the bone formation rate (BFR/BS) in the T2D mice (*Figure 2F–G*). Similarly, MS/BS and BFR/BS were both reduced at the endosteum of the bone shaft whereas MS/BS was also decreased at the periosteum (*Figure 2—figure supplement 1I–L*). These results indicate that T2D impairs osteoblast activity in both trabecular and cortical bone.

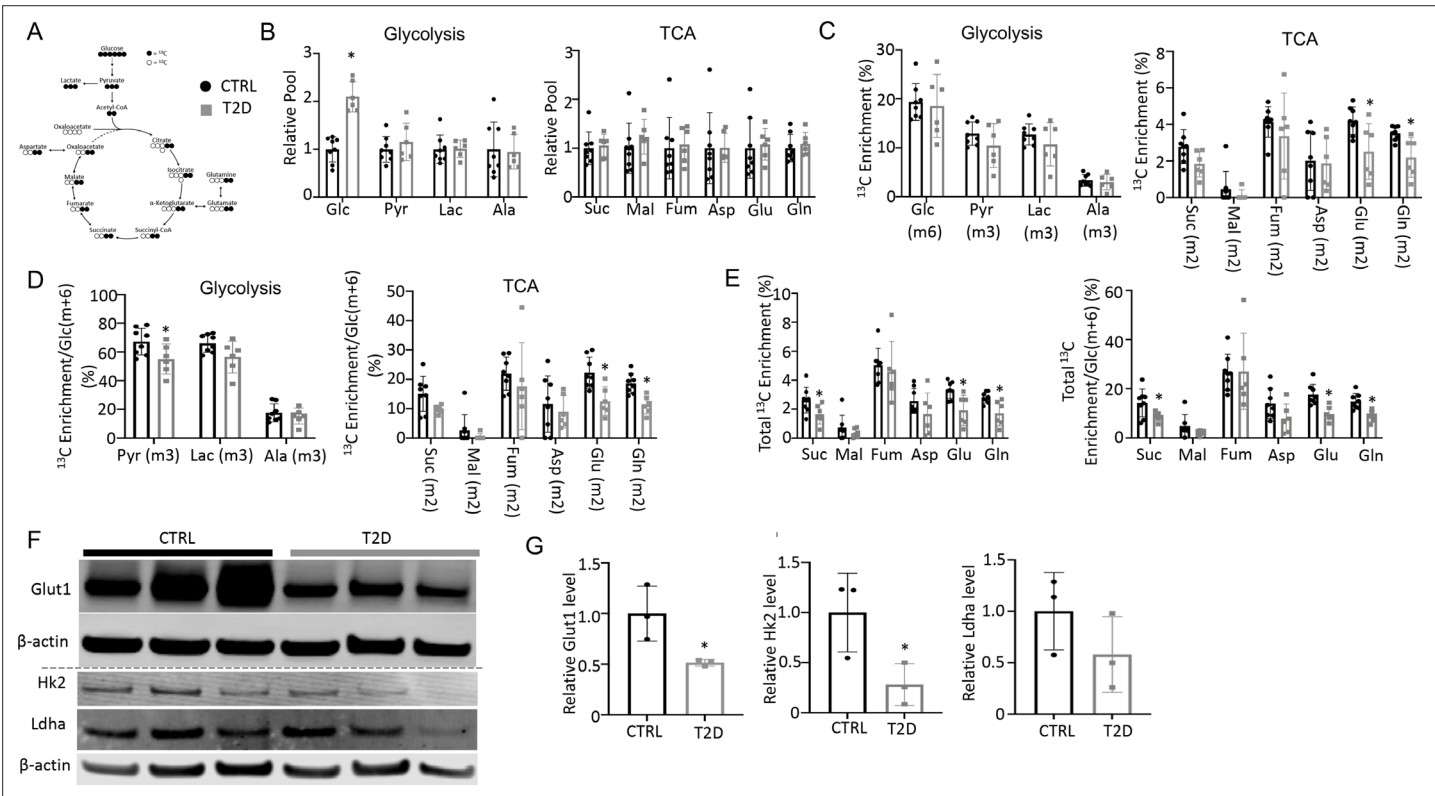

**Figure 3.** Type 2 diabetes (T2D) suppresses glucose metabolism in bone. (**A**) A diagram for carbon tracing with $^{13}C_6$-Glc. (**B**) Relative pool size of glycolytic and TCA metabolites. CTRL, n=8; T2D, n=6. (**C**) Enrichment of specific isotopologues of metabolites relative to own pool. CTRL, n=8; T2D, n=6. (**D**) Relative enrichment of specific isotopologues normalized to Glc(m+6). CTRL, n=8; T2D, n=6. (**E**) Carbon enrichment of metabolites relative to own pool (left) and normalized to Glc(m+6) (right). CTRL, n=8; T2D, n=6. (**F, G**) Western blot images (**F**) and quantifications (**G**). n=3. Data presented as mean ± SD. *p<0.05, Two-tailed unpaired t-test.

The online version of this article includes the following source data and figure supplement(s) for figure 3:

**Source data 1.** Raw images of Western blots for *Figure 3F*.

**Figure supplement 1.** Glucose tracing in plasma.

**Figure supplement 2.** Glucose tracing in bone.

## T2D impairs glucose metabolism in bone in vivo

To explore the metabolic basis for impaired osteoblast activity in T2D, we conducted stable isotope tracing in bone with uniformly labeled $^{13}$C-glucose ($^{13}C_6$-Glc). Catabolism of $^{13}C_6$-Glc was expected to produce fully labeled pyruvate Pyr(m+3) through glycolysis, which could then convert to fully labeled lactate Lac(m+3) in the cytosol or enter mitochondria to produce m+2 labeled TCA cycle metabolites in the first cycle (*Figure 3A*). For the experiment, $^{13}C_6$-Glc was injected into conscious mice (22 weeks of age) via tail vein at 60 min before the mice were harvested for detection of metabolites in bone or plasma by mass spectrometry. The glycolysis and TCA metabolite pool sizes in both bone (*Figure 3B*) and plasma (*Figure 3—figure supplement 1*) were similar between T2D and CTRL, while glucose was increased in T2D as expected. The fractional enrichment of $^{13}C_6$-Glc in bone reached approximately 20% with no difference between T2D and CTRL mice (*Figure 3C*). However, among the bone metabolites detected, the labeling enrichments of Glu(m+2) and Gln(m+2) were significantly reduced in T2D (*Figure 3C*). Moreover, when normalized to bone Glc(m+6), the relative enrichments of bone Pyr(m+3), Glu(m+2), and Gln(m+2) were decreased in T2D (*Figure 3D*). Further examination of the TCA cycle metabolites in bone revealed substantial enrichment of the m+1 isotopomers, among which Suc(m+1), Asp(m+1), Glu(m+1), and Gln(m+1) were diminished in T2D, with or without normalization to Glc(m+6) in bone (*Figure 3—figure supplement 2*). Therefore, to capture the overall contribution of $^{13}C_6$-Glc, we calculated the total carbon enrichment $((m0*0+m1*1+m2*2+....mn*n)/n)$ for each of the TCA cycle metabolites in bone. The result showed that $^{13}$C enrichments in succinate, glutamate, and glutamine, with or without normalization to bone Glc(m+6), were significantly reduced

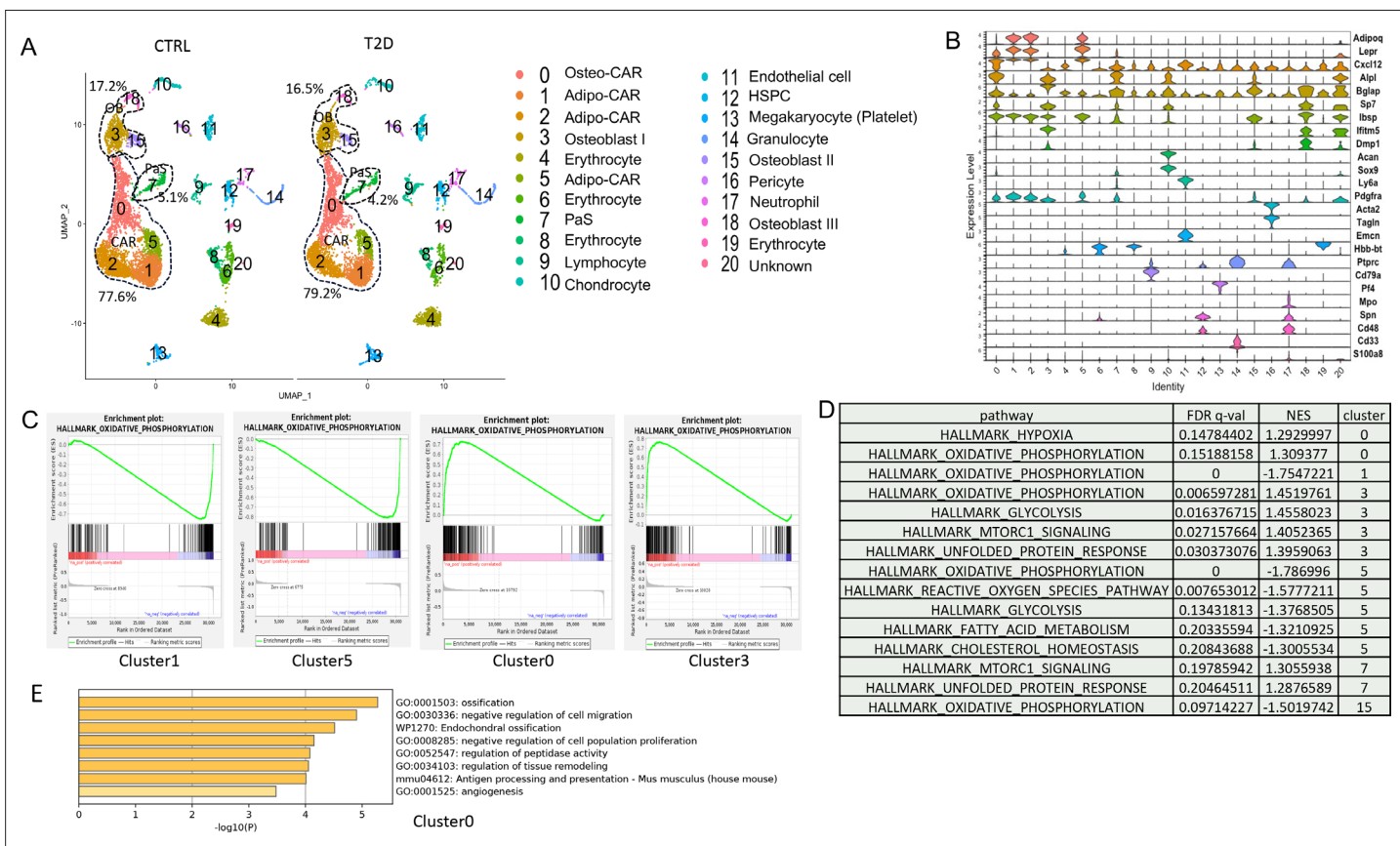

**Figure 4.** Single-cell RNA-sequencing (scRNA-seq) detects metabolic dysregulation in bone marrow mesenchymal cells of Type 2 diabetes (T2D) mice. (**A**) UMAP clusters with annotations to the right. Percentages denote the relative abundance of osteoblast (OB), CAR, and PaS cells among mesenchymal cells. (**B**) Violin plots of the example feature genes for each cluster. (**C**) Gene set enrichment analysis (GSEA) of T2D over CTRL for OXPHOS pathway in clusters as indicated. (**D**) GSEA of T2D over CTRL for metabolic pathways in clusters as indicated. (**E**) Top pathways identified among T2D downregulated genes in cluster 0 with Metascape.

The online version of this article includes the following source data for figure 4:

**Source data 1.** GSEA results for gene pathway changes between ctrl and T2D identfied by scRNA-seq.

in T2D compared to the CTRL (*Figure 3E*). The bone-intrinsic metabolic defect was further supported by Western blot analyses showing a marked decrease in Glut1 and Hk2 in the bone extracts of T2D mice (*Figure 3F and G*). Overall, T2D suppresses glucose metabolism in bone through both glycolysis and the TCA cycle.

## T2D suppresses energy metabolism and osteoblast differentiation in bone marrow mesenchymal cells

As the tracing studies above analyzed the bone shaft which contained mostly mature osteoblasts and osteocytes, we next sought to determine whether T2D similarly affected the less mature cells in the osteoblast lineage. To this end, we performed single-cell RNA-sequencing (scRNA-seq) with bone marrow mesenchymal cells purified with FACS from the bone marrow and the digested bone chips of T2D vs CTRL mice (22-week-old). Integrated analyses of the datasets uncovered 20 clusters common to both T2D and control (*Figure 4A*). Molecular annotation based on known marker genes identified a majority of the cells as *Cxcl12*-abundant reticular (CAR) cells including both Adipo-CAR (clusters 1, 2, 5) and Osteo-CAR (cluster 0) (*Baccin et al., 2020*; *Figure 4A and B*). Osteoblasts (clusters 3, 15, 18) marked by *Bglap* and *Dmp1* were also readily identified whereas cluster 7 expressed *Pdgfra* and *Ly6a* (encoding Sca1) and, therefore, were likely the PaS mesenchymal progenitors as previously reported (*Morikawa et al., 2009*). The remaining populations included chondrocytes (cluster 10), pericytes (cluster 16) (*Baryawno et al., 2019*), endothelial cells (cluster 11), erythrocytes (clusters 4, 6, 8, 19), and several hematopoietic cell types expressing *Ptprc* (encoding CD45) (clusters 9, 12, 13, 17) together with a minuscule population exhibiting both Osteo-CAR and mature osteoblast markers and, therefore, an unclear identity (cluster 20). Among CAR cells, osteoblasts, and PaS cells, the relative abundance of CAR cells was slightly increased from 77.6 to 79.2% at the expense of both osteoblasts and PaS cells. Overall, T2D does not cause major changes to the composition of bone marrow mesenchymal cells.

We next examined potential changes in gene expression levels caused by T2D within specific clusters. Gene set enrichment analysis (GSEA) detected notable effects of T2D on metabolic pathways in the mesenchymal cell clusters. Specifically, T2D suppressed the OXPHOS genes in Adipo-CAR clusters 1, 5, and osteoblast cluster 15, but had the opposite effect on Osteo-CAR cluster 0 and osteoblast cluster 3 (*Figure 4C and D*). In addition, T2D increased glycolysis genes in cluster 3 but suppressed them in cluster 5 (*Figure 4D*) (*Supplementary file 1*). It is also notable that pathways related to protein synthesis including mTORC1 signaling and unfolded protein response were increased in clusters 3 and 7 of T2D (*Figure 4D*). Further bioinformatic analyses with Metascape showed that the ossification genes were most downregulated in cluster 0 in T2D, indicating that the Osteo-CAR cells were likely suppressed in osteogenic potential (*Figure 4E*). Thus, T2D appears to disrupt normal bioenergetics and related processes in both osteoblast-lineage cells and the other mesenchymal cell types in the marrow.

To assess the actual changes in energy metabolism, we purified bone marrow stromal cells (BMSC) from the central marrow for metabolic assays in vitro. Immuno-depletion with MACS beads effectively eliminated CD45+ hematopoietic cells from the mesenchymal cells (*Figure 5—figure supplement 1*). In the T2D BMSC, both glucose consumption and lactate production were markedly reduced (*Figure 5A*). Seahorse assays detected a significant reduction in both oxygen consumption rate (OCR) and extracellular acidification rate (ECAR) in T2D BMSC (*Figure 5B and C*). The ATP production rate calculated from either glycolysis (glycoATP) or OXPHOS (mitoATP) was reduced in the T2D cells (*Figure 5D*). Thus, T2D suppresses both glycolysis and mitochondrial respiration in BMSC.

We next evaluated the potential effect of bioenergetic defects on osteoblast differentiation in T2D. We first confirmed that MACS-purified BMSC underwent robust osteoblast differentiation in vitro. BMSC from normal mice exhibited no Alizarin red staining before differentiation (day 0) but formed intensely stained nodules on day 4 and day 7 (*Figure 5—figure supplement 2A*). qPCR showed that the osteoblast makers *Ibsp*, *Bglap2*, *Alpl*, *Spp1*, *Bmp2*, *Col1a1*, *Sp7*, and *Runx2* were greatly induced after four days of differentiation, with some further upregulated on day 7 (*Figure 5—figure supplement 2B*). Importantly, when cultured in parallel with the normal cells, T2D BMSC expressed markedly lower levels of virtually all osteoblast markers examined on both day 4 and day 7 of differentiation (*Figure 5E and F*). The differentiation defect was further supported by reduced Alizarin red staining of the minerals after 4 or 7 days of differentiation (*Figure 5G*). Overall,

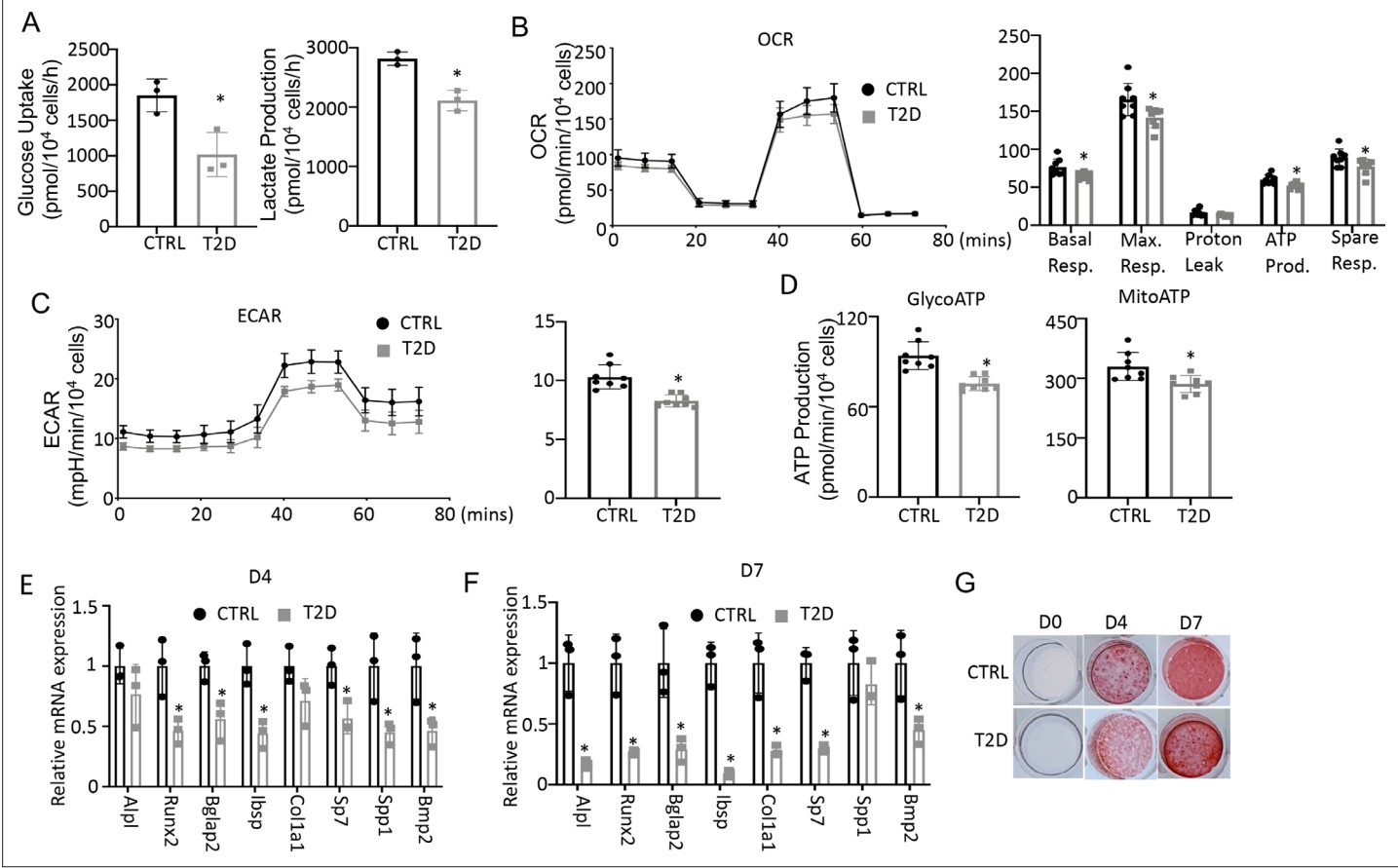

**Figure 5.** Type 2 diabetes (T2D) impairs glucose metabolism and osteogenic differentiation in bone marrow stromal cells (BMSC). (**A**) Glucose uptake and lactate production rates. n=3. (**B, C**) Seahorse measurements of oxygen consumption rate (OCR) (**B**) and extracellular acidification rate (ECAR) (**C**). n=8. (**D**) Projected ATP production glycolysis (glycoATP) and mitochondria (mitoATP) based on Seahorse. n=8. (**E, F**) qPCR for osteoblast markers at day 4 (**E**) and day 7 (**F**) of differentiation. n=3. (**G**) Alizarin red staining. Data presented as mean ± SD. *p<0.05, Unpaired student's t-test.

The online version of this article includes the following figure supplement(s) for figure 5:

**Figure supplement 1.** Bone marrow stromal cells (BMSC) purification with MACS beads.

**Figure supplement 2.** In vitro osteoblast differentiation of MACS-purified bone marrow stromal cells (BMSC) from wild-type mice.

the data indicate that T2D causes cell-intrinsic defects in both metabolic and osteogenic properties of BMSC.

## Enhancing glycolysis with metformin ameliorates bone loss in T2D

To test if impaired glucose utilization drives osteopenia in T2D, we sought to boost glycolysis and examine its effect on bone mass. Metformin is a widely prescribed anti-diabetic and has been proposed to boost glycolysis through a variety of mechanisms (*Moonira et al., 2020*; *Valero, 2014*; *Wang et al., 2019*; *Zhu et al., 2020*). We, therefore, administered metformin to the T2D mice through drinking water and examined the effects on metabolic and bone parameters (*Figure 6A*). GTT and ITT showed that glucose tolerance and insulin sensitivity were modestly improved by the metformin regimen although fasting glucose levels remained higher than normal (*Figure 6B–E*). Metformin did not affect body weight, body composition, or circulating insulin levels in the T2D mice (*Figure 6—figure supplement 1A–C*). Importantly, μCT showed that metformin notably increased trabecular bone mass (BV/TV) mainly due to increased Tb. N and Conn. Dens coupled with decreased Tb. Sp, with no changes in trabecular thickness or cortical bone parameters (*Figure 6F*, *Figure 6—figure supplement 1D, E*). Double labeling revealed that metformin increased mineralizing surfaces (MS/BS) without altering the mineral apposition rate (MAR), resulting in a 50% increase in bone formation rate

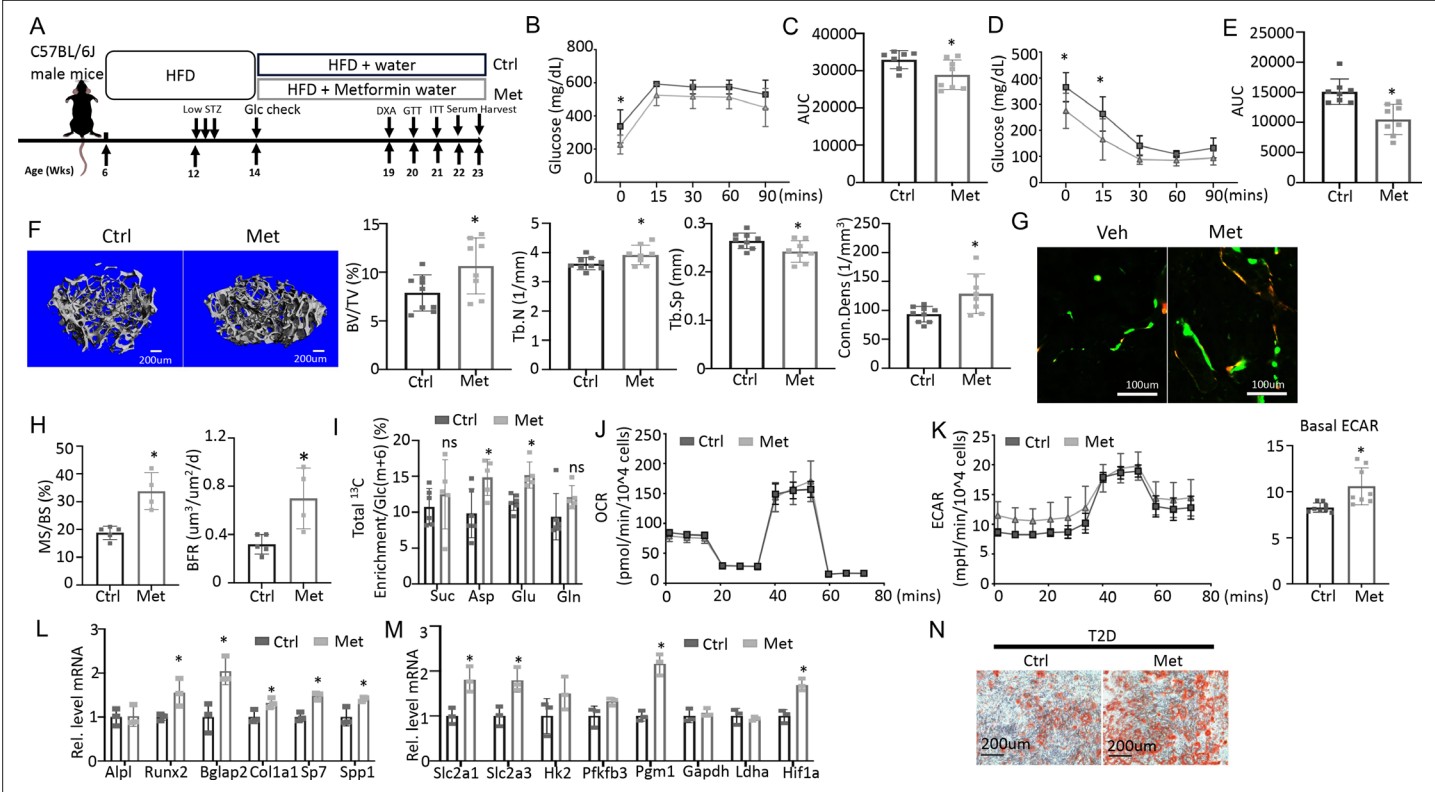

**Figure 6.** Metformin improves bone mass and glucose metabolism impaired by Type 2 diabetes (T2D). (**A**) A schematic for metformin regimen. T2D animals were randomly divided into Control (Ctrl) and metformin (Met) groups. (**B, C**) Glucose tolerance test (GTT) curve (**B**) and area under curve (**C**). Ctrl, n=7; Met, n=8. (**D, E**) ITT curve (**D**) and area under curve (**E**). n=8. (**F**) Representative μCT images and quantification of trabecular bone in the distal femur. Ctrl, n=9; Met, n=8. (**G, H**) Representative images (**G**) and quantification (**H**) of double labeling in trabecular bone of the distal femur. Ctrl, n=5; Met, n=4. (**I**) Carbon enrichment of TCA metabolites normalized to Glc(m+6). Ctrl, n=6; Met, n=5. (**J, K**) Seahorse measurements of oxygen consumption rate (OCR) (**J**) and extracellular acidification rate (ECAR) (**K**) with or without 1 mM metformin treatment for 24 hr. n=8. (**L, M**) qPCR analyses of osteoblast markers (**L**) and glycolysis-related genes (**M**) in bone marrow stromal cells (BMSC) at day 7 of differentiation with or without 1 mM metformin treatment. n=3. (**N**) Alizarin red staining at day 7 of osteoblast differentiation. Data are presented as mean ± SD. *p<0.05, two-way ANOVA followed by Sidak's multiple comparisons (**B, D**) or Student's t-test (all others).

The online version of this article includes the following figure supplement(s) for figure 6:

**Figure supplement 1.** Metformin improves bone mass in Type 2 diabetes (T2D) mice.

(BFR) in the treated T2D mice (**Figure 6G and H**, **Figure 6—figure supplement 1F**). Thus, metformin mitigates bone loss in T2D by promoting bone formation.

To assess the potential effect of metformin on glucose metabolism in bone, we next performed stable isotope tracing with $^{13}C_6$-Glc in the T2D mice with or without metformin treatment immediately before harvest at 23 weeks of age (**Figure 6A**). Both Pyr(m+3) and Lac(m+3) normalized to Glc(m+6) in bone exhibited a trend of increase in response to metformin, but the differences did not achieve statistical significance, likely due to insufficient power of the sample size (**Figure 6—figure supplement 1G**, p=0.05). On the other hand, the total carbon enrichment of aspartate and glutamate relative to Glc(m+6) was significantly increased by metformin, indicating increased glucose entry into the TCA cycle in bone (**Figure 6I**). The data, therefore, support that metformin promotes bone glucose metabolism in T2D mice.

We next examined whether metformin has a direct effect on osteoblast-lineage cells. For metabolic effects, BMSC from T2D mice were treated with metformin for 24 hr before Seahorse assays. Metformin had a negligible effect on OCR but significantly elevated ECAR (**Figure 6J and K**). Moreover, in osteoblast differentiation assays, metformin increased the expression of multiple osteoblast markers and key glycolysis genes in T2D BMSC (**Figure 6L and M**). Similarly, metformin notably improved the formation of mineralized nodules by the T2D cells following osteoblast differentiation

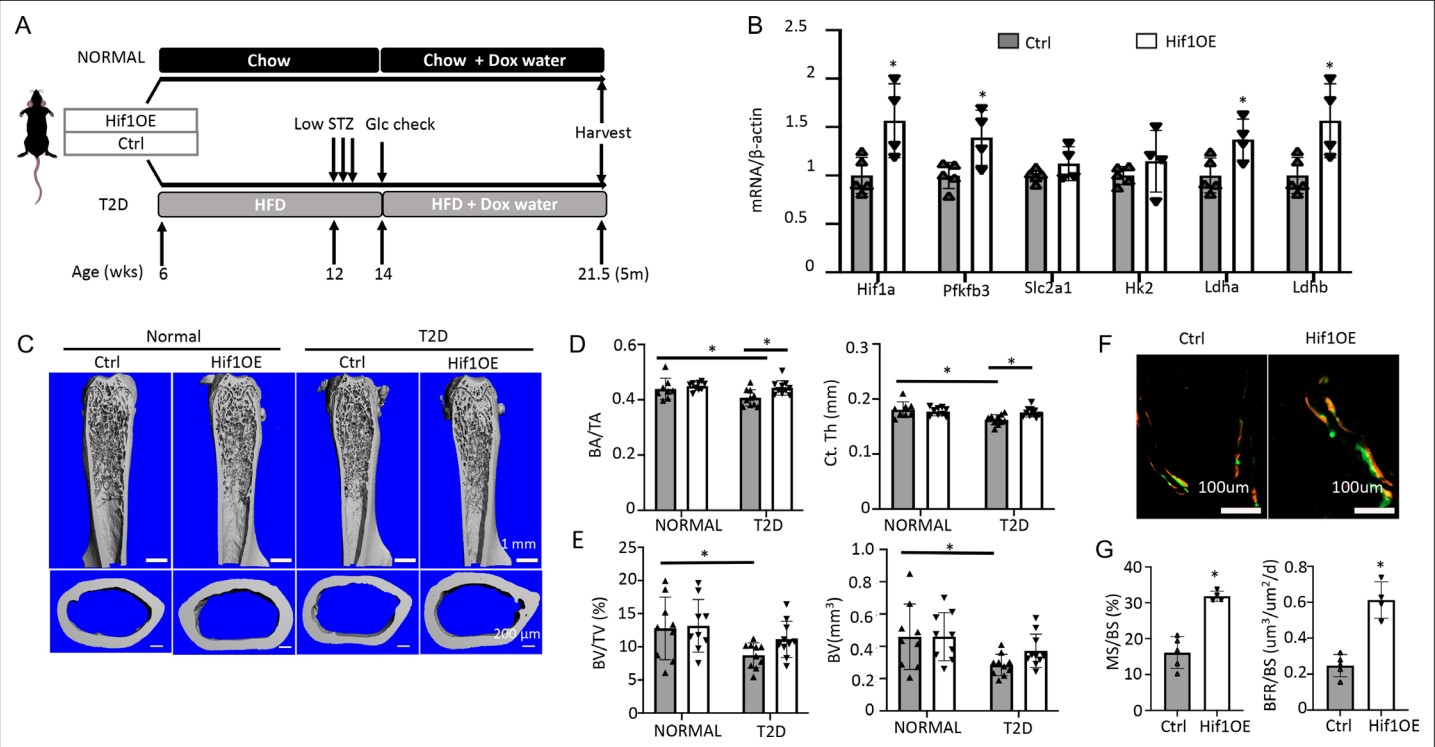

**Figure 7.** Targeted overexpression of Hif1a in osteoblasts rescues low bone mass in Type 2 diabetes (T2D) mice. (**A**) A schematic for Hif1a overexpression (Hif1OE) in normal or T2D conditions. (**B**) qPCR analyses of glycolysis-related genes in bones of diabetic mice with or without Hif1a overexpression. n=4. (**C–E**) Representative µCT images (**C**) and quantification of cortical bone (**D**) and distal trabecular bone (**E**) of the femur. n=10. (**F**, **G**) Representative images (**F**) and quantification (**G**) of double labeling in trabecular bone of proximal tibia in diabetic mice. Ctrl, n=5, Hif1OE, n=4. Data presented as mean ± SD. *p<0.05 Unpaired Student's t-test (**B, G**) or two-way ANOVA followed by Fisher's LSD test (**D, E**).

The online version of this article includes the following source data and figure supplement(s) for figure 7:

**Figure supplement 1.** Hif1a overexpression improves bone mass in Type 2 diabetes (T2D) mice.

**Figure supplement 2.** Glut1 overexpression does not improve bone in Type 2 diabetes (T2D) mice.

**Figure supplement 2—source data 1.** Raw images of Western blots for *Figure 7—figure supplement 2A*.

**Figure supplement 2—source data 2.** Raw western blots.

(*Figure 6N*). Together, these findings indicate that metformin directly enhances both glycolysis and osteoblast differentiation in T2D BMSC.

## Genetic activation of glycolysis reduces bone loss in T2D

To corroborate the direct effect on bone, we utilized a transgenic mouse model that increased glycolysis specifically in the osteoblast-lineage cells. The transcription factor Hif1α is known to stimulate the expression of numerous genes in the glycolytic pathway, thus promoting glycolysis. Here, we generated mice with the genotype of Osx-rtTA; tetO-Cre; Hif1dPA (Hif1OE hereafter) to allow for Cre-mediated expression of a degradation-resistant form of Hif1a (Hif1dPA) in the osteoblast lineage upon Dox administration. Their sex-matched littermates missing at least one of the three transgenes were used as controls (Ctrl). Both Hif1OE and Ctrl males were either fed the regular chow (normal group) or induced to develop T2D (*Figure 7A*). Dox was supplied in the drinking water for both groups at 14 weeks of age until harvest at 21.5 weeks. Gene expression studies with bones from the T2D group confirmed that Hif1a and several glycolysis-related genes including Pfkfb3, Ldha, and Ldhb, were upregulated in Hif1OE over Ctrl mice (*Figure 7B*). In the normal group, µCT analyses of the cortical bone detected no change in cortical bone fraction (BA/TA) or cortical thickness (Ct. Th), although there was an unexpected decrease in the cross-sectional size (TA and BA) in Hif1OE (*Figure 7C and D*, *Figure 7—figure supplement 1A*). Importantly, under T2D conditions, Hif1OE showed a significant increase in both cortical bone fraction (BA/TA) and cortical thickness (Ct. Th) when compared to Ctrl

(*Figure 7D*). No difference was detected in the trabecular bone parameters of Hif1OE and Ctrl in the normal group (*Figure 7E*, *Figure 7—figure supplement 1B*). However, among the T2D mice, trabecular bone faction (BV/TV) and volume (BV) trended higher in Hif1OE than Ctrl mice, without changes in the other parameters (*Figure 7E*, *Figure 7—figure supplement 1B*). Student's t-test of BV/TV or BV between the two diabetic groups detected statistical significance ($p<0.05$), but two-way ANOVA among all four groups did not confirm a significant effect by Hif1OE on the trabecular bone, likely due to insufficient power of the sample size. Double labeling experiments in the diabetic mice detected a notable increase in trabecular bone formation (BFR/BS), thanks to increased MS/BS but unchanged MAR in Hif1OE (*Figure 7F and G*, *Figure 7—figure supplement 1C*). Overall, Hif1a overexpression in osteoblast-lineage cells stimulates bone formation in the T2D but not the healthy mice.

We next explored specific steps in the glycolytic pathway that may be targeted to enhance bone formation in the T2D mice. Overexpression of Glut1, a major glucose transporter in osteoblasts, has been shown to promote bone formation in normal mice (*Wei et al., 2015*). We, therefore, generated mice with the genotype of Osx-rtTA; TRE-Glut1 (Glut1OE) to induce Glut1 overexpression in osteoblast-lineage cells with Dox. Both Glut1OE males and their sex-matched littermates (Ctrl) missing one of the two transgenes were rendered diabetic, and then exposed to Dox according to the same regimen as described for the Hif1OE mice. Western blot analyses demonstrated successful overexpression of Glut1 in the bone shaft of long bones in Glut1OE (*Figure 7—figure supplement 2A*). However, μCT analyses detected no significant change in either trabecular or cortical bone parameters between Glut1OE and Ctrl mice (*Figure 7—figure supplement 2B, C*). Likewise, serum P1NP and CTX-1 levels were similar between the genotypes, with P1NP trending down in the Glut1OE mice

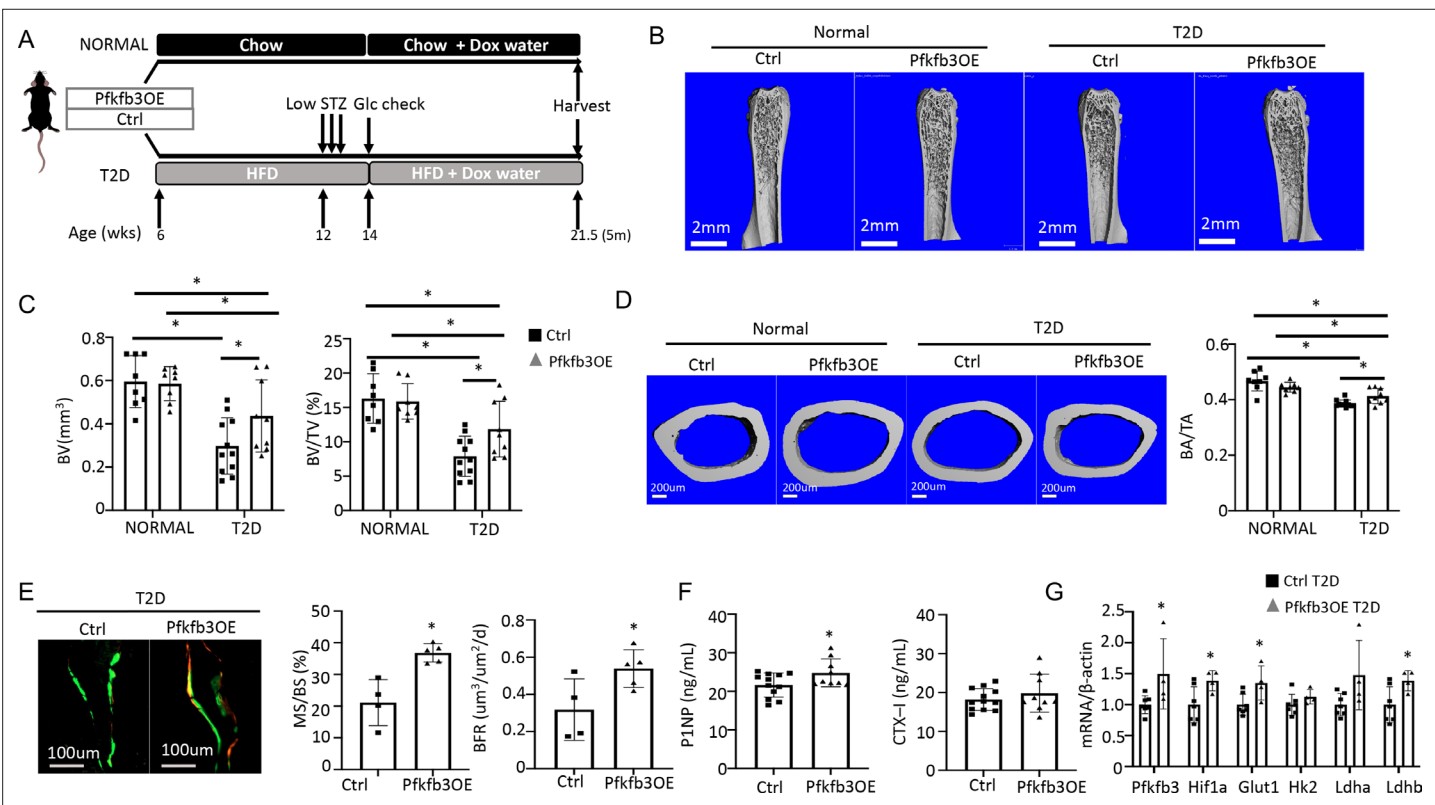

**Figure 8.** Osteoblast-directed Pfkfb3 overexpression corrects bone loss in Type 2 diabetes (T2D). (**A**) A schematic for experimental design. Representative μCT images (**B**) and quantification (**C**) of trabecular bone in the distal femur. (**D**) Representative μCT images and quantification of cortical bone in femur. Ctrl NORMAL, n = 8; Pfkfb3OE NORMAL, n = 8; Ctrl T2D, n = 11; Pfkfb3OE T2D, n = 9. (**E**) Representative images and quantification of double labeling in trabecular bone of the proximal tibia. Ctrl T2D, n = 4; Pfkfb3OE T2D, n = 5. (**F**) Serum markers for bone formation (P1NP) and resorption (CTX-1). Ctrl T2D, n = 12; Pfkfb3OE T2D, n = 9. (**G**) qPCR analyses of glycolysis-related genes in bone. Ctrl T2D, n = 7; Pfkfb3OE T2D n = 4. Data presented as mean ± SD. *$p<0.05$, Two-way ANOVA followed by Fisher's LSD test (**C, D**) or Student's t-test (all others).

The online version of this article includes the following figure supplement(s) for figure 8:

**Figure supplement 1.** Pfkfb3 overexpression improves bone mass in Type 2 diabetes (T2D mice).

(*Figure 7—figure supplement 2D*). Thus, increased Glut1 expression does not improve osteopenia in T2D mice.

We next investigate the potential effect of phosphofructokinase-2/fructose-2,6-bisphosphatase (Pfkfb3) which functions downstream of Glut1 in the core glycolysis pathway. Pfkfb3 synthesizes fructose-2,6-bisphosphate (F2,6P2), a potent allosteric activator of 6-phosphofructo-1-kinase (Pfk1) which in turn controls a rate-limiting step of glycolysis (*De Bock et al., 2013*). In a parallel study to be published elsewhere, we found that overexpression of Pfkfb3 in primary cultures of murine calvarial preosteoblasts markedly stimulated glucose consumption and lactate production. We further generated a mouse strain (R26-TRE-Pfkfb3, or TRE-Pfkfb3 in short) to allow for overexpression of Pfkfb3 in vivo in a doxycycline-dependent manner. Here, we generated Osx-rtTA; TRE-Pfkfb3 mice (Pfkfb3OE) along with their control littermates (Ctrl) missing at least one of the two transgenes. Cohorts of Pfkfb3OE and Ctrl mice were each divided into a normal or T2D group. In the T2D group, the mice were rendered diabetic and then treated with Dox for the remainder of the experiment, whereas those in the normal group were maintained on regular chow before being treated with Dox in the same way (*Figure 8A*). Upon harvest of the mice, µCT analyses detected no difference in any of the trabecular bone parameters between Pfkfb3OE and Ctrl mice in the normal group (*Figure 8B*, *Figure 8—figure supplement 1A*). However, within the T2D group, Pfkfb3OE significantly increased trabecular BV, trabecular bone fraction (BV/TV), coupled with increased Conn. Dens, increased Tb. N and reduced Tb. Sp (*Figure 8B and C*, *Figure 8—figure supplement 1A*). Similarly, Pfkfb3 overexpression did not change the cortical bone parameters in the normal group, but it increased cortical bone fraction (BA/TA) in the T2D group, thanks to the correction of total cross-sectional area (TA) (*Figure 8D*, *Figure 8—figure supplement 1B*). Dynamic histomorphometry revealed that Pfkfb3 overexpression in T2D mice increased BFR due to a marked increase in bone MS/BS without affecting the MAR (*Figure 8E*, *Figure 8—figure supplement 1C*). Serum biochemical assays showed that Pfkfb3 overexpression elevated P1NP levels without changing CTX-I in the T2D mice (*Figure 8F*). RT-qPCR detected a modest increase of Pfkfb3, Hif1a, Glut1, and Ldhb mRNA in the bone shaft of Pfkfb3OE diabetic mice versus Ctrl T2D mice (*Figure 8G*). Overall, the results demonstrate that bone-specific activation of glycolysis is an effective anabolic strategy to counter the bone loss in T2D mice.

## Discussion

We have investigated the mechanism underlying the bone anabolic defect in a T2D mouse model. Evidence from multiple approaches including stable isotope tracing in vivo, scRNA-seq of bone marrow mesenchymal cells, as well as in vitro flux assays supports that T2D causes metabolic dysregulation intrinsic to osteoblast lineage cells. Importantly, activation of glycolysis either pharmacologically or by genetic means effectively enhances bone formation and preserves bone mass in diabetic mice. The results, therefore, support osteoblast-intrinsic impairment of glucose metabolism as a pathogenic mechanism for diabetic osteopenia in T2D.

Although adult-onset T2D differs from youth-onset T2D in that the net bone mass is either normal or even increased in adults, they both suppress bone formation. Multiple mechanisms including hyperglycemia and decreased signaling by Wnt, adipokine, insulin, or Igf1 have been proposed to impair osteoblast differentiation and function in diabetes, but their functional relevance in vivo remains elusive (*Napoli et al., 2017*). Nonetheless, as Wnt and Igf1 signaling has been shown to promote glycolysis directly in osteoblast lineage cells, it is reasonable to speculate that the downregulation of the aforementioned signals may be a proximate cause for impaired glycolysis in diabetic bone (*Esen et al., 2013*; *Esen et al., 2015*). Our mouse model recapitulates the low bone turnover as seen in human patients, but we have chosen to focus on the bone anabolic defect as it drives the osteopenia phenotype. The model can be further studied in the future to determine the mechanism for impaired bone resorption in T2D and its potential contribution to impaired bone formation through a coupling mechanism. It is worth noting that others have previously reported increased osteoclasts and bone resorption in HFD-fed mice (*Shu et al., 2015*). Thus, the osteoclast defect observed here is caused by diabetes instead of HFD feeding per se. In any case, the current study shows that osteoblast glycolysis can be selectively targeted to increase bone mass without causing 'coupled' activation of bone resorption in T2D.

We have focused on glucose metabolism here, but hyperlipidemia is expected to occur in the T2D mice not only because of the HFD feeding but also because hyperglycemia and hyperinsulinemia can

increase lipid accumulation in tissues (*Winhofer et al., 2012*). Previous studies have linked hyperlipidemia with bone loss in both humans and rodents, and have further indicated that the effect is partly mediated through suppression of osteoblast differentiation (*Tintut and Demer, 2014*). However, the intracellular mechanism mediating the anti-osteogenic effect of lipids has not been elucidated. Our finding that activation of glycolysis can restore bone formation in the face of hyperlipidemia argues that impaired glycolysis could be a major driver for the bone pathology downstream of hyperlipidemia. It should be interesting then to test whether hyperlipidemia suppresses glycolysis in osteoblasts in the future. The potential effect could be caused by either bioactive lipids signaling through putative receptors, or interruption of normal fatty acid metabolism previously implicated in osteoblast bioenergetics (*Tintut and Demer, 2014*; *Kim et al., 2017*).

Our study shows that stimulation of glycolysis either pharmacologically or by genetic activation promotes bone formation in T2D. In keeping with its bone anabolic function in vivo, metformin increased the mRNA levels of multiple genes directly or indirectly involved in glycolysis, including Slc2a1, Slc2a3, Pgm1, and Hif1a, during osteoblast differentiation in vitro. However, as metformin also improved the overall diabetic phenotype in the T2D mice, the rescue of osteoblasts could potentially be an indirect effect. Thus, targeted activation of glycolysis in osteoblasts was necessary to demonstrate a direct effect in vivo. While Hif1a is well documented to stimulate glycolysis, its target genes are numerous and include Vegf which could induce angiogenesis to secondarily promote bone formation. Pfkfb3 overexpression, on the other hand, is presumed to target a single step of glycolysis and, therefore, provides more definitive evidence for a cell-autonomous role of increased glycolysis in osteoblasts.

Overexpression of either Hif1a or Pfkfb3 did not affect bone formation in the normal mice, indicating that osteoblast glycolysis is unlikely to be rate-limiting for bone homeostasis in healthy adult mice. We previously showed that overexpression of a different form of stabilized Hif1a in osteoblasts increased trabecular bone mass in young healthy mice (*Regan et al., 2014*). However, there are important differences between the two studies. In the earlier study, Hif1a-PPN carrying three alanine substitutions (P402A, P564A, and N803A) was expressed from a tetO promoter in the transgene in response to Dox. Here, Hif1dPA (P402A, P564A) was expressed from the Rosa26 endogenous locus upon removal of a stop cassette by Cre in response to Dox. The differences in promoter strength and the point mutations could lead to variable activity levels for Hif1a in osteoblasts between the two models. Moreover, the earlier study analyzed the bone mass at 5 weeks of age when the mice were actively accruing bone, but the current analysis was performed with 5-month-old adult mice which had a considerably lower basal bone formation rate.

Overexpression of Glut1 failed to rescue bone formation, indicating that glucose uptake was not responsible for the impaired bone anabolic activity in T2D. The result further supports the notion that the downregulation of Glut1 in diabetic bones likely represents a protective mechanism against cellular damage, as an intracellular accumulation of glucose is known to be cytotoxic through multiple mechanisms (*Brownlee, 2001*). In particular, activation of the polyol pathway could deplete both NADPH and $NAD^+$ in osteoblasts. Whereas NADPH is critical for glutathione homeostasis and oxidative stress control, $NAD^+$ is required as a co-factor for glycolysis. On the other hand, hyperactivation of the hexosamine biosynthetic pathway could impair protein functions via abnormal glycosylation. It's important to note that both pathways branch off the core glycolysis pathway upstream of the step stimulated by Pfkfb3. Thus, Pfkfb3 overexpression may diminish the flux into the harmful side pathways while boosting core glycolysis in osteoblasts of the T2D mice. Future experiments are necessary to determine the potential flux changes in vivo.

In conclusion, the study identifies impaired glucose metabolism in osteoblasts as a key mediator for the bone anabolic defect in T2D. Furthermore, osteoblast glycolysis offers a promising target for developing future bone therapies.

## Materials and methods
### Mice
Animal experiments were conducted with approval from the Institutional Animal Use and Care Committee at Children's Hospital of Philadelphia (IACUC NO: IAC 21–001296). Mice were housed with a 12:12 hr light-dark cycle at 22°C. The TRE-Pfkfb3 mouse was generated by placing a

doxycycline-inducible Pfkfb3 (CDS isoform 6, NCBI NM_133232.3) expression cassette into the Rosa26 locus by homologous recombination. Chimeric founder mice were created by blastocyst injection of the correctly targeted ES cells and then mated with C57BL/6 N to confirm germ-line transmission. The other mouse strains including TRE-Glut1, Osx-rtTA, Hif1dPA, and TetO-Cre were as previously described, and maintained on the C57BL/6 J background (*Song et al., 2020*; *Pereira et al., 2013*; *Kim et al., 2006*; *Perl et al., 2002*). For Dox-mediated gene induction, Dox was supplied in drinking water at 1 mg/ml (Doxycycline hyclate, D9891, Sigma).

## T2D mouse model

For T2D induction, C57BL/6 J male mice were fed ad libitum a high-fat diet (HFD, Research diet D12492, 60 kcal% Fat) starting at 6 weeks of age, injected with low-dose streptozotocin (STZ, 30 mg/kg, Sigma) once daily for three consecutive days (5 hr fasting before each injection) starting at 12 weeks of age, and continued on HFD until harvest at 22 weeks of age. Males were used because they were more prone to develop T2D with the protocol. The control group was C57BL/6 J male mice of the same age and fed ad libitum regular chow (CTRL, Rodent Laboratory Chow 5015, Purina, USA). Both groups had free access to water. Random glucose level was measured at two weeks after the first STZ injection, and mice with levels lower than 200 mg/dL were excluded from further study. Metformin hydrochloride (4 mg/mL, Cayman Chemical, No. 13118) and Dox (1 mg/mL) were dissolved in drinking water with the water bottle changed twice weekly.

## GTT and ITT

Glucose tolerance test (GTT) and insulin tolerance test (ITT) were carried out with one-week intervals for the animals to recover from stress. For ITT, mice were injected intraperitoneally with human insulin at 0.5 U/kg body weight (Novalin R U-100) after 6 hr of fasting. Blood samples were collected at the indicated time (0, 15, 30, 60, and 90 min after injection) from a cut in the tail for glucose measurements with a glucometer. For GTT, mice were intraperitoneally injected with glucose at 2 g/kg body weight after 6 hr of fasting. Blood glucose levels were measured at 0, 15, 30, 60, 90, and 120 min after injection. The upper limit of the glucometer was 600 mg/dL; glucose levels were recorded as 600 mg/dL when the reading was out of range.

## Serum biochemical assays

Sera were collected from mice through retro-orbital bleeding after 6 hr of fasting. CTX-I and P1NP assays were performed with the RatLaps (CTX-I) EIA and Rat/Mouse P1NP EIA Kit (Immunodiagnostic Systems, Ltd.), respectively. Insulin and Igf1 assays were performed with a mouse insulin ELISA kit (Ultra Sensitive Mouse Insulin ELISA, Cat#90080, Crystal Chem) and a mouse Igf1 ELISA kit (Mouse IGF-1 ELISA Kit, Cat#80574, Crystal Chem). HOMA-IR was calculated as follows: HOMA IR = fasting insulin (mU/L) * fasting glucose (mg/dL)/405. Fasting glucose levels were measured with a glucometer in blood collected from a tail-vein cut.

## Skeletal phenotyping

The distal end of the femur was scanned by µCT (µCT45; SCANCO Medical) at 4.5 µm isotropic voxel size. For trabecular bone analyses, regions of interest were selected at 0.45–2.25 mm below the growth plate. For cortical bone, a total of 70 slices at the femur midshaft located at 5.4 mm away from the distal growth plate were analyzed. Parameters computed from these data included BV, TV, BV/TV, Tb.Th, Tb.N, Tb. Sp, and Conn. Dens at the distal femoral metaphysis and TA, BA, BA/TA and Ct. th at the mid-diaphysis of the femur.

Dynamic histomorphometry was conducted by sequential injections of calcein green (5 mg/kg body weight (bw)) and alizarin red (15 mg/kg bw) at 7 and 2 days, respectively, prior to sacrifice. Femurs or tibias were fixed with 4% PFA in PBS for 48 hr and then kept in 70% ethanol with protection from light; the fixed bones were switched to 30% sucrose in PBS overnight before OCT embedding and cryostat sectioning. Longitudinal 10 µm cryostat sections of the undecalcified bones were collected with Cryofilm type II membrane (Section Lab, Co. Ltd., Japan) and imaged using Axioscan (ZEISS, Germany). Quantification was performed with BIOQUANT Image Analysis System (BIOQUANT Image Analysis Corporation, USA).

## Stable isotope tracing in vivo

Conscious animals were fasted for 6 hr and then injected with one bolus of *Lee et al., 2020* [13]$C_6$-Glc (40 mg/mouse) via tail vein at 60 min before the plasma and bone shafts were harvested *Lee et al., 2020*. For plasma metabolite extraction, plasma (5 µl) was added to 120 µl of −20°C 40:40:20 methanol: acetonitrile: water (extraction solvent), vortexed, and centrifuged at 21,000 g for 20 min at 4°C. The supernatant (40 µl) was collected for liquid chromatography–mass spectrometry (LC–MS) analysis. For bone metabolite extraction, frozen bones (1 tibia and 1 femur) were transferred to 2 ml round-bottom Eppendorf Safe-Lock tubes on dry ice. Samples were then ground into a powder with a cryoMill machine (Retsch) for 30s at 25 Hz, and maintained at a cold temperature using liquid nitrogen. For every 20 mg tissue, 800 µl of −20°C extraction solvent was added to the tube, vortexed for 10s, and then centrifuged at 21,000 × g for 20 min at 4°C. The supernatants (500 µl) were transferred to another Eppendorf tube, dried using an N2 evaporator (Organomation Associates, Berlin, MA), and redissolved in 50 µl of −20°C extraction solvent. Redissolved samples were centrifuged at 21,000 × g for another 20 min at 4°C. The supernatants were then transferred to plastic vials for LC-MS analysis. A procedure blank was generated identically without tissue, and was used later to remove the background ions.

For metabolite measurements by LC-MS, a quadrupole-orbitrap mass spectrometer (Q Exactive Plus, Thermo Fisher Scientific, San Jose, CA) operating in negative mode was coupled to hydrophobic interaction chromatography (HILIC) via electrospray ionization. Scans were performed from m/z 70–1000 at 1 Hz and 140,000 resolution. LC separation was conducted on an XBridge BEH Amide column (2.1 mm × 150 mm × 2.5 mm particle size, 130 Å pore size; Water, Milford, MA) using a gradient of solvent A (20 mM ammonium acetate, 20 mM ammonium hydroxide in 95:5 water: acetonitrile, pH 9.45) and solvent B (acetonitrile). Flow rate was 150 mL/min. The LC gradient was: 0 min, 85% B; 2 min, 85% B; 3 min, 80% B; 5 min, 80% B; 6 min, 75% B; 7 min, 75% B; 8 min, 70% B; 9 min, 70% B; 10 min, 50% B; 12 min, 50% B; 13 min, 25% B; 16 min, 25% B; 18 min, 0% B; 23 min, 0% B; 24 min, 85% B. Autosampler temperature was 5 °C, and injection volume was 15 µL. For improved detection of fructose-1,6-bisphosphate and 3-phosphoglycerate, selected ion monitoring (SIM) scans were added. For 3-phosphoglycerate and fructose-1,6-bisphosphate, scans were performed from 180 to 190 m/z and 336–350 m/z, respectively, from 13 to 15 min of the 25 min gradient run with 70,000 resolution and maximum IT of 500 ms and AGC target of $3 \times 10^6$. Data were analyzed using the El-MAVEN (Elucidata) software *Melamud et al., 2010*. Carbon Enrichment was calculated as follows: Carbon Enrichment = (m0*0+m1*1+m2*2+….mn*n)/n.

## scRNA-seq

For isolation of bone marrow mesenchymal cells, epiphyseal ends of both tibias and femurs were removed and the bone marrow cells were flushed out with M-199 (Thermo Fisher Scientific, USA) supplemented with 10% FBS (heat-inactivated Qualified FBS, Thermo Fisher Scientific, USA) and collected. The bone shaft was then cut into pieces and digested with 1 mg/mL STEMxyme1 (Worthington, Ref#LS004106) and 1 mg/mL Dispase II (ThermoFisher Scientific, Ref#17105041), in M-199 supplemented with 2% FBS for 25 min at 37 °C. The digested cells and bone marrow cells were pooled together and passed through a 70 mm filter. Cells were then stained in media 199 supplemented with 2% FBS for FACS. The antibodies and staining reagents included Ter119-APC (eBioscience, Ref#17-5921-82, clone TER-119), CD71-PECy7 (Biolegend, Ref#113812, clone RI7217), CD45-PE (eBioscience, Ref#12-0451-82, clone 30-F11), CD3-PE (Biolegend, Ref#100206, clone 17A2), B220-PE (Biolegend, Ref#103208, clone RA3- 6B2), CD19-PE (Biolegend, Ref#115508, clone 6D5), Gr-1-PE (Biolegend, Ref#108408, clone RB6-8C5), Cd11b-PE (Biolegend, Ref#101208, clone M1/70), and Calcein AM (Thermo Fisher Scientific, Massachusetts, USA) for 30 min on ice. 7-AAD (Thermo Fisher Scientific, Massachusetts, USA) was added into the medium before sorting on MoFlo Astrios (Beckman Coulter, Brea, CA, USA). Bone marrow mesenchymal cells were enriched as the 7-AAD[-]/ Calcein AM[+]/ CD71[-]/ Ter119[-]/ Lin (CD45/CD3/B220/CD19/Gr-1/CD11b)[-] fraction.

scRNA-seq libraries were constructed with bone marrow mesenchymal cells from T2D vs CTRL mice with the Single Cell 3' Reagent Kits v3.1 (10x Genomics). The cells were pooled from three T2D or CTRL mice. Sequencing was performed with a NovaSeq sequencer using 100-cycle paired-end reads, generating 200 million reads per sample. Cell ranger (10x Genomics) was used for demultiplexing, extraction of cell barcode, and quantification of unique molecular identifiers (UMIs). 8608

and 8629 cells were recovered from CTRL and T2D samples, respectively. Downstream analyses were performed using Seurat v3 *Stuart et al., 2019*. Cells with poor quality (genes >6000,, genes <200, or >10% mitochondrial genes) were excluded. We then normalized the data by the total expression, multiplied by a scale factor of 10,000, and log-transformed the result. The medium number of features detected per cell was 2088 and 2333 from CTRL and T2D, respectively. For the integrated datasets, anchors were defined using the FindIntegrationAnchors function, and these anchors from the different datasets were then used to integrate datasets together with IntegrateData. Different resolutions were used to demonstrate the robustness of clusters. In addition, differentially expressed genes within each cluster between two datasets were identified using FindAllMarkers. GSEA with MsigDB gene sets was performed to identify pathways with significant changes. Metascape web tool was used to analyze DEG lists between T2D and Ctrl, with parameters as follows: Min Overlap: 3, p-Value Cutoff: 0.01, Min Enrichment: 1.5.

## BMSC isolation and purification

Tibias and femurs were dissected clean from adjacent tissues, and the epiphyses containing the growth plates were excised off with scissors. The bone marrow cells were flushed out with a syringe filled with MEM-α media containing 10% FBS and then passed through a cell strainer (35 μm). The cells for all four bones of one mouse were seeded into one 100 mm dish. The culture medium (MEM-α containing 10% FBS) was changed every day after 4 days following seeding, and the cells normally became confluent at day 6–8, at which time they were dissociated using Collagenase II (4 mg/ml) and 0.05% trypsin with EDTA for 5 min at 37 °C. Hematopoietic lineage cells were immuno-depleted using the MACS technique (Miltenyi Biotec, USA). Briefly, the cells were mixed with CD45 antibody-coated magnetic beads (CD45 MicroBeads, 130-052-301) and then passed through a MACS LD column (Miltenyi Biotec, 130-042-901) on a MidiMACS separator (Miltenyi Biotec, 130-042-302) attached to a multistand. The purified cells were cultured and passaged once for osteogenic differentiation and metabolic assays.

For osteoblast differentiation, when the BMSCs reached 100% confluency, MEM-α containing 4 mM ß-glycerol phosphate (Sigma, G9422), and 50 ug/ml ascorbic acid (Sigma, A4544) was added and then changed daily. Cells after 4 or 7 days of differentiation were dissociated with 4 mg/ml collagenase I in PBS for 45 min, then with 0.05% trypsin for 10 min for RNA extraction or metabolic assays.

## In vitro metabolic assays

For Seahorse assays, BMSC purified with MACS were seeded at $2 \times 10^5$ cells/cm$^2$ into XF96 tissue culture microplates (Agilent) at 24 hr prior to experiments. Complete Seahorse medium was prepared from Agilent Seahorse XF Base Medium (Agilent, 102353) containing 5.5 mM glucose, 2 mM glutamine, and 1 mM pyruvate, with pH 7.4. The cells were incubated in 180 μl complete Seahorse medium at 37 °C for 1 hr in a CO$_2$-free incubator before measurements in Seahorse XFe96 Analyzer. The following working concentrations of compounds were used: 2 μM Oligomycin, 3 μM FCCP, 1 μM Rotenone, and 1 μM Antimycin A. The oxygen consumption rate (OCR) and extracellular acidification rate (ECAR) were normalized to the seeded cell number. Theoretical ATP production rates from glycolysis and OXPHOS was calculated based on the reading of OCR and ECAR. Wave Pro software was used to calculate the following parameters: Proton efflux rate (PER), glycoPER, glycoATP Production Rate, mitoATP Production Rate, and TotalATP Production Rate.

For glucose consumption and lactate production, BMSC were seeded at $4 \times 10^5$ cells/cm$^2$ in six-well plates and cultured in a customized medium (with glucose, pyruvate, and glutamine reconstituted before use) for 24 hr. The medium was collected for glucose or lactate measurement with Glucose (HK) Assay Kit (Sigma, GAHK20) or L-Lactate Assay Kit I (Eton Bioscience, 120001), respectively. The cells were dissociated and counted for normalization.

## RT-qPCR

Total RNA was harvested from cells or bone shafts using an RNeasy mini kit (QIAGEN) according to the manufacturer's protocol. 1 μg RNA was reverse transcribed into cDNA using High-Capacity cDNA Reverse Transcription Kit (Thermo Fisher Scientific, Massachusetts, USA). The gene expression level relative to β-actin was determined by qPCR with gene-specific primers (*Supplementary file 2*) and

PowerUp SYBR Green Master Mix (Thermo Fisher Scientific, USA). $2^{-\Delta\Delta CT}$ was calculated for relative fold change.

## Western blot analyses

Protein extracts from bone were prepared in M-PER buffer (cat#78501, Thermo Fisher Scientific, USA) containing phosphatase and proteinase inhibitors (cat#78442, Halt Protease and Phosphatase Inhibitor Single-Use Cocktail, Thermo Fisher Scientific, USA). After surgical removal of the epiphysis and flushing of the marrow, the bone shafts of femurs and tibias were snap-frozen in liquid nitrogen and then homogenized with a Precellys homogenizer (Bertin Instrument, France). Proteins were separated by electrophoresis in NuPAGE 4 to 12% Bis-Tris Mini Protein Gels (Thermo Fisher Scientific, USA), and then transferred to Immobilon-FL PVDF Membrane (Millipore, USA). The membranes were blocked with Odyssey Blocking Buffer (PBS) (LI-COR Biosciences, USA), and then incubated with primary antibodies overnight. The primary antibodies included Anti-Glucose Transporter Glut1 antibody [EPR3915] (ab115730, 1:2000), Anti-Hexokinase II antibody [EPR20839] (ab209847, 1:1000), Anti-Lactate Dehydrogenase antibody [EP1566Y] (ab52488, 1:1000) and Anti-beta Actin antibody [AC-15] (ab6276, 1:2000). The secondary antibodies IRDye 800CW Donkey anti-Rabbit IgG (H+L) (LI-COR Biosciences, 1: 20,000) and IRDye 680RD Goat anti-Mouse IgG (H+L) (LI-COR Biosciences, 1:20,000) were used. The protein bands were imaged with Odyssey DLx Imaging System and quantified with Image J.

## Statistics

Statistics were conducted with unpaired Student's t-test, one-way ANOVA followed by Student's t-test between multiple groups or two-way ANOVA with either Sidak's multiple comparisons or Fisher's LSD test, as indicated in the figure legends, by using Prism Software 9.0. p-value <0.05 is considered significant. All quantitative data are presented as mean ± SD.

## Acknowledgements

The work is partially supported by NIH grant R01 DK125498 (FL).

## Additional information

### Funding

| Funder | Grant reference number | Author |
| --- | --- | --- |
| National Institutes of Health | DK125498 | Fanxin Long |

The funders had no role in study design, data collection and interpretation, or the decision to submit the work for publication.

### Author contributions

Fangfang Song, Data curation, Formal analysis, Investigation, Methodology, Writing – original draft; Won Dong Lee, Methodology, Writing - review and editing; Tyler Marmo, Investigation, Writing - review and editing; Xing Ji, Chao Song, Xueyang Liao, Rebecca Seeley, Investigation; Lutian Yao, Haoran Liu, Software, Visualization; Fanxin Long, Conceptualization, Formal analysis, Supervision, Funding acquisition, Writing - review and editing

### Author ORCIDs

Fangfang Song (ID) http://orcid.org/0000-0001-9465-6083
Won Dong Lee (ID) http://orcid.org/0000-0002-2344-171X
Fanxin Long (ID) http://orcid.org/0000-0001-9785-5379

### Ethics

All animal studies were conducted in strict accordance with the protocols approved by the Institutional Animal Care and Use Committee (IACUC) at Children's Hospital of Philadelphia. The IACUC approval number is IAC 21-001296.

Decision letter and Author response
Decision letter https://doi.org/10.7554/eLife.85714.sa1
Author response https://doi.org/10.7554/eLife.85714.sa2

## Additional files

### Supplementary files

• Supplementary file 1. GSEA analysis of gene expression changes in each cell cluster identified by scRNA-seq (T2D vs CTRL, FDR q<0.25).
• Supplementary file 2. Nucleotide sequences of qPCR primers.
• MDAR checklist

### Data availability

scRNA-seq data has been deposited in GEO under access code GSE221936All source data for figures and the R script for scRNA analyses have been uploaded to Dryad (https://doi.org/10.5061/dryad.s7h44j1bc).

The following datasets were generated:

| Author(s) | Year | Dataset title | Dataset URL | Database and Identifier |
|---|---|---|---|---|
| Song F, Long F | 2023 | Osteoblast-intrinsic defect in glucose metabolism impairs bone formation in type II diabetic male mice | http://www.ncbi.nlm.nih.gov/geo/query/acc.cgi?acc=GSE221936 | NCBI Gene Expression Omnibus, GSE221936 |
| Long F | 2023 | Data from: Osteoblast-intrinsic defect in glucose metabolism impairs bone formation in type II diabetic male mice | http://dx.doi.org/10.5061/dryad.s7h44j1bc | Dryad Digital Repository, 10.5061/dryad.s7h44j1bc |

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
