## [Editor Report]

The authors have established an important mouse model for youth-onset T2D and provided convincing evidence to demonstrate that suppression of glucose metabolism in osteoblasts is the main cause of impaired bone formation in T2D. Particularly, the authors uncovered that activation of glycolysis in osteoblasts restores bone formation in T2D mice as an essential mechanism. This study fundamentally demonstrated osteoblast-intrinsic defects in glucose metabolism as an underlying cause of diabetic osteopenia with therapeutical potential.

---

## [Decision Letter]

**Decision letter after peer review:**

Thank you for submitting your article "Osteoblast-intrinsic defect in glucose metabolism impairs bone formation in type II diabetic mice" for consideration by *eLife*. Your article has been reviewed by 3 peer reviewers, and the evaluation has been overseen by a Reviewing Editor and Mone Zaidi as the Senior Editor. The reviewers have opted to remain anonymous.

Essential revisions:

1) The authors use two approaches: a drug (Metformin) and a number of mouse genetic models to over-express genes involved in the glycolytic pathway using Dox inducible models. The results with overexpressing HIF1 and PFKFB3 show a potential rescue of bone defects with T2D, and Glut1 overexpression does not rescue T2D-induced bone loss.

2) HFD-fed obese and T2D mice had more periosteal reactive bone, osteolysis, and osteoclast activation. For example, the increase of osteoclast activity has been previously reported in HFD-fed mice which leads to bone loss and alters bone metabolism.

*Reviewer #1 (Recommendations for the authors):*

The paper is generally excellent with an interesting scientific premise, that the effects of T2DM on bone could be mediated through osteoblast metabolic dysfunction and particularly dysfunction in glucose uptake, especially since it is already known that bone is partly responsible for glucose tolerance and a relative lack of response to insulin in T2DM patients. The approach of using a mouse model is good, although the low-dose STZ model seems less relevant to the clinic than (e.g.) a high-fat diet or other model that more closely recapitulates the way T2DM is formed in humans. However, it isn't a bad choice overall. Overall, the paper is generally strong and only needs minor changes in general.

– The second figure needs some sort of checks to see if osteoblast numbers are affected by the T2DM model. This could be done using staining for ALP and counting osteoblast numbers per surface area of the bone.

– The fourth figure is very short on material and could use considerable expansion here and in/or in the supplemental information. Specifically, more direct analysis of osteoblast, and other bone cells (e.g. mesenchymal cell) expression of Glut transporter levels. Further, a more thorough GSEA analysis for changed pathways may be provided, especially showing the most enriched pathways with differentially expressed genes in the T2D model.

– The fifth figure needs confirmation of the alizarin red staining and qPCR for osteoblast activity with two methods – there needs to be staining with Von Kossa staining for osteoblast activity measurement as well as western blotting for changes in gene expression.

*Reviewer #2 (Recommendations for the authors):*

I don't have any significant comments here.

*Reviewer #3 (Recommendations for the authors):*

1) The investigators used HFD to induce T2D and discussed the impact of glucose metabolism on osteoblast activity and osteopenia. However, lipid metabolism is also important to the osteoblast activity and is relevant in this animal model but was neither studied nor discussed.

2) HFD-fed obese and T2D mice had more periosteal reactive bone, osteolysis, and osteoclast activation. For example, the increase of osteoclast activity has been previously reported in HFD-fed mice which leads to bone loss and alters bone metabolism. (https://www.frontiersin.org/articles/10.3389/fcell.2022.832460/full). The serum CTX didn't change in the current study. Were there any changes in osteoclasts in this model?

3) Figure 6 showed several glycolysis-related gene expressions increased by metformin including Hif1a. It is interesting to further explore the role of Hif1a in metformin's effects on BMSC glycolysis. Whether metformin effects are abolished in the absence of Hif1a needs to be studied. The in vivo result of Hif1a OE mice is not conclusive in Figure 7 as the baseline data of WT and HifOE mice fed on a normal diet is lacking.

4) Pfkfb3 is not regulated by metformin thus the result in Figure 8 on the Pfkfb3OE mice needs to be discussed in the context of the whole study.

---

## [Author Response]

Essential revisions:1) The authors use two approaches: a drug (Metformin) and a number of mouse genetic models to over-express genes involved in the glycolytic pathway using Dox inducible models. The results with overexpressing HIF1 and PFKFB3 show a potential rescue of bone defects with T2D, and Glut1 overexpression does not rescue T2D-induced bone loss.

The editor raises an important question as to why increased glucose uptake in osteoblasts fails to rescue diabetic bone loss. Hyperglycemia in diabetes is known to cause pathological activation of the polyol pathway and the Hexosamine Biosynthetic Pathway (HBP) inside the cell, both of which branch off from the core glycolysis pathway upstream of the step stimulated by Pfkfb3. Indeed, our scRNA-seq data from the bone marrow cells show that Gfpt1, encoding the first and rate-limiting enzyme for HBP, is upregulated by T2D in multiple clusters including CAR cells (clusters #0, 1, 2) and osteoblasts (#3), whereas the paralogous gene Gfpt2 is less affected (Author response image 1). Therefore, increasing glucose uptake in osteoblasts by Glut1 overexpression may exacerbate the polyol and HBP pathways already activated in diabetes. Hif1a is known to activate multiple key enzymes throughout the glycolysis pathway. Pfkfb3 overexpression stimulates a step downstream of the branch-off point of either the polyol pathway or HBP. Therefore, overexpression of either Hif1a or Pfkfb3 likely diminishes the detrimental side pathways by boosting the core glycolytic flux.

**Author response image 1. sa2fig1:** scRNA-seq detects increased expression of Gfpt1 in multiple mesenchymal cell clusters in the bone marrow of T2D mice compared to control. Violin plots show relative gene expression levels in each cell cluster (x-axis) in control (red) versus T2D (green) mice.

2) HFD-fed obese and T2D mice had more periosteal reactive bone, osteolysis, and osteoclast activation. For example, the increase of osteoclast activity has been previously reported in HFD-fed mice which leads to bone loss and alters bone metabolism.

A previous study reported that obesity and T2D increases periosteal reaction and osteolysis during *Staphylococcus aureus* implant-associated bone infection (Farnsworth et al., 2018, J Orthop Res 36(6): 1614-23). Those effects however were the results of heightened inflammation in response to the bacterial infection, and therefore not relevant to this study which deals with bone homeostasis. In fact, our study shows that bone resorption as reflected by serum CTX-I levels is reduced in the T2D mice.

Reviewer #1 (Recommendations for the authors):The paper is generally excellent with an interesting scientific premise, that the effects of T2DM on bone could be mediated through osteoblast metabolic dysfunction and particularly dysfunction in glucose uptake, especially since it is already known that bone is partly responsible for glucose tolerance and a relative lack of response to insulin in T2DM patients. The approach of using a mouse model is good, although the low-dose STZ model seems less relevant to the clinic than (e.g.) a high-fat diet or other model that more closely recapitulates the way T2DM is formed in humans. However, it isn't a bad choice overall. Overall, the paper is generally strong and only needs minor changes in general.

We thank the reviewer for the positive comments. We acknowledge that the use of STZ is irrelevant to the etiology of T2D in humans. However, because HFD alone was generally not sufficient to induce a frank T2D phenotype in the mouse, we adopted the method of combining HFD feeding with low dose TZD, which, as we demonstrated in the paper, was effective in this regard.

– The second figure needs some sort of checks to see if osteoblast numbers are affected by the T2DM model. This could be done using staining for ALP and counting osteoblast numbers per surface area of the bone.

We agree that static histomorphometry based on cell morphology or ALP staining (marking both immature and mature osteoblasts) could reveal overall changes in the osteoblast lineage, but they do not distinguish active bone-forming osteoblasts from precursors or inactive cells. We therefore have relied on dynamic histomorphometry to provide an integrated assessment of active osteoblast number and function.

– The fourth figure is very short on material and could use considerable expansion here and in/or in the supplemental information. Specifically, more direct analysis of osteoblast, and other bone cells (e.g. mesenchymal cell) expression of Glut transporter levels. Further, a more thorough GSEA analysis for changed pathways may be provided, especially showing the most enriched pathways with differentially expressed genes in the T2D model.

Thanks for the suggestion. Further analyses of osteoblast and other mesenchymal cells did not reveal any significant changes in the expression levels of Glut transporters. We have now expanded this figure to include additional results from pathway analyses.

– The fifth figure needs confirmation of the alizarin red staining and qPCR for osteoblast activity with two methods – there needs to be staining with Von Kossa staining for osteoblast activity measurement as well as western blotting for changes in gene expression.

We appreciate the thoroughness of the reviewer but feel that the data from qPCR and alizarin red staining, together with the in vivo findings, already provide strong evidence that osteoblast differentiation is impaired in the diabetic mouse.

Reviewer #3 (Recommendations for the authors):1) The investigators used HFD to induce T2D and discussed the impact of glucose metabolism on osteoblast activity and osteopenia. However, lipid metabolism is also important to the osteoblast activity and is relevant in this animal model but was neither studied nor discussed.

This is an excellent point. Hyperlipidemia is expected in the T2D mice because of the HFD feeding. Moreover, hyperglycemia and hyperinsulinemia may increase lipid accumulation in bone as it has been shown in the heart (PMC3331780). Previous studies have linked hyperlipidemia with bone loss in both humans and rodents, and have further indicated that the effect is partly mediated through suppression of osteoblast differentiation (PMC3946677). However, the intracellular mechanism mediating the anti-osteogenic effect of lipids has not been elucidated. Our finding that activation of glycolysis can restore bone formation in the face of hyperlipidemia argues that impaired glycolysis is a major driver for the bone pathology. It should be interesting then to test whether hyperlipidemia suppresses glycolysis in osteoblasts in the future. The potential effect could be caused by either bioactive lipids which has been postulated to signal through putative receptors, or interruption of normal fatty acid metabolism previously shown to participate in energy production in osteoblasts (PMC3946677, PMC5621897). We now include these comments in Discussion.

2) HFD-fed obese and T2D mice had more periosteal reactive bone, osteolysis, and osteoclast activation. For example, the increase of osteoclast activity has been previously reported in HFD-fed mice which leads to bone loss and alters bone metabolism. (https://www.frontiersin.org/articles/10.3389/fcell.2022.832460/full). The serum CTX didn't change in the current study. Were there any changes in osteoclasts in this model?

In our T2D model, the serum CTX level was reduced (Figure 2E), indicating suppression of osteoclast bone resorption. The different results about bone resorption between this and the previous study as cited by the reviewer could be explained by the fact that the previous study used HFD alone which did not lead to overt T2D.

3) Figure 6 showed several glycolysis-related gene expressions increased by metformin including Hif1a. It is interesting to further explore the role of Hif1a in metformin's effects on BMSC glycolysis. Whether metformin effects are abolished in the absence of Hif1a needs to be studied. The in vivo result of Hif1a OE mice is not conclusive in Figure 7 as the baseline data of WT and HifOE mice fed on a normal diet is lacking.

The reviewer raises an excellent point that metformin might stimulate glycolysis through Hif1a in BMSC; we plan to investigate this possibility in the future. We now provide the baseline data of WT and HifOE mice on a normal diet and show no difference in cortical or trabecular bone mass (Figure 7). Thus, the rescue effect of Hif1aOE is specific to the T2D bones, indicating that osteoblast glycolysis becomes rate limiting for bone formation in T2D.

4) Pfkfb3 is not regulated by metformin thus the result in Figure 8 on the Pfkfb3OE mice needs to be discussed in the context of the whole study.

We did not examine whether Pfkfb3 is regulated by metformin in BMSC, but others have shown that metformin stimulates Pfkfb3 expression in adipocytes (PMC5512603). Regardless, in the current study we target Pfkfb3 because it stimulates a specific step of glycolysis (conversion of fructose-6-phosphate to fructose 1, 6-bisphosphate). The findings from the Pfkfb3OE mice further buttress the conclusion that boosting glycolysis is sufficient to improve bone formation in the T2D mice. We have expanded the discussion to integrate the results from both metformin and the genetic studies.